# Frailty and risk of cardiovascular disease and mortality

Xiao Liu[1], Nien Xiang Tou[1], Qi Gao[2], Xinyi Gwee[2], Shiou Liang Wee[1,3], Tze Pin Ng[1,2]*

1 Geriatric Education and Research Institute, Singapore, Singapore, 2 Gerontology Research Programme, Department of Psychological Medicine, Yong Loo Lin School of Medicine, National University of Singapore, Singapore, Singapore, 3 Health and Social Sciences Cluster, Singapore Institute of Technology, Singapore, Singapore

* pcmngtp@nus.edu.sg

## Abstract

### Background

Prospective cohort studies suggest that frailty is associated with an increased risk of incident cardiovascular disease (CVD) morbidity and mortality, but their mechanistic and developmental relations are not fully understood. We investigated whether frailty predicted an increased risk of incident nonfatal and fatal CVD among community-dwelling older adults.

### Methods

A population cohort of 5015 participants aged 55 years and above free of CVD at baseline was followed for up to 10 years. Pre-frailty and frailty were defined as the presence of 1–2 and 3–5 modified Fried criteria (unintentional weight loss, weakness, slow gait speed, exhaustion, and low physical activity), incident CVD events as newly diagnosed registered cases of myocardial infarction (MI), stroke, and CVD-related mortality (ICD 9: 390 to 459 or ICD-10: I00 to I99). Covariate measures included traditional cardio-metabolic and vascular risk factors, medication therapies, Geriatric Depression Scale (GDS), Mini-Mental State Exam (MMSE), and blood biomarkers (haemoglobin, albumin, white blood cell counts and creatinine).

### Results

Pre-frailty and frailty were significantly associated with elevated HR = 1.26 (95%CI: 1.02–1.56) and HR = 1.54 (95%CI:1.00–2.35) of overall CVD, adjusted for cardio-metabolic and vascular risk factors and medication therapies, but not after adjustment for GDS depression and MMSE cognitive impairment. The HR of association between frailty status and both CVD mortality and overall mortality, however, remained significantly elevated after full adjustment for depression, cognitive and blood biomarkers.

### Conclusion

Frailty was associated with increased risk of CVD morbidity and especially mortality, mediated in parts by traditional cardio-metabolic and vascular risk factors, and co-morbid

**Data Availability Statement:** All relevant data are within the manuscript.

**Funding:** The study was supported by research grants from the Agency for Science Technology and Research (A*STAR) Biomedical Research Council (BMRC) (https://www.a-star.edu.sg/)

[Grant: 08/1/21/19/567] and from the National Medical Research Council (https://www.nmrc.gov.sg/) [Grant: NMRC/1108/2007]. NTP received both fundings. The funders had no role in study design, data collection and analysis, decision to publish, or preparation of the manuscript.

**Competing interests:** The authors have declared that no competing interests exist.

depression and associated cognitive impairment and chronic inflammation. Given that pre-frailty and frailty are reversible by multi-domain lifestyle and health interventions, there is potential benefits in reducing cardiovascular diseases burden and mortality from interventions targeting pre-frailty and early frailty population.

## Introduction

Frailty is a common geriatric syndrome reflecting a state of reduced physiological reserve and increased vulnerability to the effects of stress [1]. The population prevalence of frailty and pre-frailty, defined using the Fried criteria is high, estimated at 17.4% and 49.3% respectively [2]. Frailty occurs in as much as 50% of older patients with cardiovascular disease (CVD) [3]. Both frail and pre-frail individuals compared to their robust counterparts have higher likelihoods of presenting with CVD, and vice versa [4]. Results from cross-sectional study supported an independent association between subclinical vasculopathy with muscle mass and strength, determinants of frailty [5]. However cross-sectional associations are unable to establish the causal relationship between the frailty and CVD. On the one hand, CVD has been shown to be an important predictor of the onset of frailty, and the presence of frailty in older adults with CVD increases the risk of falls, institutionalization, repeated hospitalization, and mortality [6–8]. On the other hand, frailty has been suggested as a risk factor for the development of CVD [9–11]. However, few prospective cohort studies [12–14] have investigated whether pre-frailty and frailty predict an earlier onset of CVD events and mortality.

Previous studies have reported the association of pre-frailty/frailty with incident hospitalization for heart failure and for overall CVD events, and not separately for non-fatal and fatal CVD events. Physical inactivity and slow gait (in one study [13]) and exhaustion (in two studies [13, 14]) were found to be significantly associated with the onset of CVD events. Depression is associated with cognitive impairment and both are well established comorbidities of frailty [15]. In turn, depression and inflammatory biomarkers are associated with CVD incidence [16]. The role of depression in explaining the association of frailty and CVD risk has not been investigated. The mechanistic and developmental relationships between frailty and CVD risk therefore remains not fully understood.

As frailty might be reversible if appropriately treated [17, 18], the timely detection and therapeutic interventions for frailty and the precursor pre-frailty may have a positive impact in terms of postponing or preventing onset of coronary heart disease, heart failure, stroke and mortality in older persons. The aim of the present prospective cohort study was to investigate the association of pre-frailty and frailty with the risk of developing CVD morbidity and mortality from 10-years follow-up in a cohort of community-dwelling older adults in an Asian population in Singapore.

## Methods

### Study population

The Singapore Longitudinal Ageing Study (SLAS) is a population-based study that recruited community-dwelling older adults (age>55) who were able to self-ambulate and with adequate cognitive capacity for participation in two separate recruitment waves. SLAS-1 recruited 2,800 older persons in the South-East Region in 2003–2004, and SLAS-2 recruited 3,200 individuals in the South Central and Western Region in Singapore in 2009–2013, each with 3 to 5 yearly follow-ups. The SLAS was approved by the National University of Singapore Institutional

Review Board with all participants consented by written form. Full details of the study variables and data collection are described in previous studies [19, 20].

## Study sample

In this study, we excluded participants with a confirmed diagnosis of acute myocardial infarction (MI) (n = 38) and stroke (n = 59) at baseline, subjects with self-reported history of atrial fibrillation, heart attack, and heart failure (n = 715) at baseline, and subjects with missing data on frailty and frailty components (n = 296). Our final sample size was 5,015, combining 2,426 participants from SLAS 1 and 2,589 from SLAS 2.

## Measurements

All-cause mortality and fatal CVD cases were obtained from the Death Registry data from Singapore National Registry of Diseases Office based on International Classification of Diseases (ICD). Fatal CVDs were identified using ICD 9 codes from 390 to 459 or ICD 10 codes from I00 to I99. Other CVD outcomes included 1) non-fatal MI, obtained from Singapore Myocardial Infarction Registry; 2) non-fatal stroke, obtained from Singapore Stroke Registry; 3) non-fatal CVD, defined as an inclusion of non-fatal MI and non-fatal stroke. Overall CVD included both fatal CVD and non-fatal CVD. Overall mortality includes all-cause of death cases. The follow-up time for this study started at the date of participants enrolment and ended in December 2017 for all the outcomes.

Frailty was defined according to Fried's five criteria in the Cardiovascular Health Study [1]. Each domain (Shrinking, Low activity, Weakness, Exhaustion, Slowness) accounted for 1 point, and participants were categorized as frail (3–5 points), prefrail (1–2 points), or robust (0 point) based on the sum of all five items. The detailed frailty measurements were described in previous study [19] and summarized below.

1. Shrinking or weight loss: body mass index (BMI) of less than 18.5 kg/m2 and/or unintentional weight loss of ≥4.5 kg (10 pounds) in the past 6 months.

2. Weakness was defined as the lowest quintile of knee extension strength within sex and BMI strata in SLAS-2 participants. In SLAS-1 participants, this was defined as the lowest quintile of score of rising from chair test in the sitting position with arms folded, derived from the Performance Oriented Mobility Assessment (POMA) battery [21].

3. Slowness was defined as gait speed less than 0.8m/s from the fast gait speed test over 6 metres in SLAS2 participants. In SLAS2 participants, slowness was assessed by Tinetti POMA gait tests (subjects walked 6 meters and returned to the starting point quickly), which include 7 gait items—initiation of gait, step length and height, step symmetry, step continuity, path, trunk and walking stance. The total POMA gait score has a range from 0 to 12, and a score of less than 9 denotes impaired gait functioning.

4. Exhaustion was determined by the response of "not at all" to the question from SF-12 quality of life scale: "Do you have a lot of energy?"

5. Low activity was determined by self-report of "none" for participation in any physical activity (walking or recreational or sports activity).

One-point was assigned for the presence of each component, and the total score categorizes participants as frail (3–5 points), pre-frail (1–2 points), or robust (0 point).

## Baseline covariates

Sociodemographic information included age, sex (male versus female), race (Chinese versus Non-Chinese), education (no education, 1–6 years primary and post-primary), housing (1–2 room, 3–5 room and high-end public/private housing), marital status (married versus single/divorced/widowed) and living status (alone versus not alone). Lifestyle behavior included smoking (current smoker versus non-smoke) and alcohol use (daily drinker versus non-daily drinker). Depressive symptoms were assessed by the Geriatric Depression Scale (GDS) score ≥5. Mini-Mental State Examination (MMSE) was used to categorize participants as cognitive impaired (MMSE score <24). Metabolic syndrome was defined according to the International Diabetes Federation including central obesity, raised triglycerides (TG), reduced high-density lipoprotein cholesterol (HDL-C), hypertension and diabetes [22]. Raised low-density lipoprotein cholesterol (LDL-C) was defined as ≥3.4mmol/l [23]. Medication therapies included statin therapy, antiplatelet therapy, anticoagulant therapy. Other blood biomarkers contained hemoglobin (g/L), albumin (g/L), creatinine (umol/L) and white blood cell (WBC) (x10^9/L).

## Statistical analysis

The analyses used means (SD) for continuous variables and proportions (N) for categorical variables of frailty, frailty domains, and covariates at baseline in the overall sample, and compared CVD versus non-CVD outcomes using two-sample t-tests and chi-square tests for significance tests. Hierarchical adjusted Cox proportional hazard models were used to estimate hazard ratios (HR) and their 95% confidence intervals (95% CIs) between frailty status and overall incident CVD, and between frailty status and overall mortality. Competing-risks survival regression models were performed to estimate sub-distribution hazard ratios (SHR) and their 95% CI between frailty status and frailty domains and other CVD outcomes described above. HR of incident CVD for frail versus robust, and prefrail versus robust were estimated first in the unadjusted Cox proportional hazard model. Covariates were included in Models 1 to 5 in sequential hierarchical order. Model 1: additionally adjusted for age and sex; Model 2: additionally for socio-demographics (race, education, housing); Model 3: additionally for smoking, alcohol, central obesity, raised TG, reduced HDL-C, diabetes, hypertension, raised LDL-C, statin therapy, antiplatelet therapy, anticoagulant therapy; Model 4: additionally, for GDS depression and MMSE; Model 5: additionally for blood biomarkers. The "time to event" was defined by the length of time between baseline and the first recorded CVD event. Sensitivity analysis excluding CVD cases within 1 year after baseline was performed. A two-sided p value of 0.05 was considered as statistically significant. All analysis was performed using Stata 13.0 (Stata Corporation, College Station, TX, USA).

## Results

The mean age of the overall sample was 65.8 (SD = 7.6); nearly two-thirds were female (65.2%); and the majority were Chinese (90.6%); 19% were without an education; and 13.4% were living in lower-end 1–2 room apartments. In all, 3.7% of the participants were frail and nearly half (46.2%) were pre-frail. The prevalence of frailty domains was 26.4% for "Low activity", 18.1% for "Weakness", 11.4% for "Exhaustion", 8.6% for "Shrinking", and 4.5% for "Slowness".

As shown in Table 1, participants with CVD events compared to those without differed significantly on baseline characteristics of frailty and frailty-related risk factors, showing higher baseline frequencies of pre-frailty and frailty and frailty domains, indexes of socioeconomic deprivation and isolation, depression and cognitive impairment, as well as established cardio-

**Table 1. Baseline characteristics in overall sample and by CVD and non-CVD outcomes.**

| Baseline characteristics | Whole sample (n = 5015) | | CVD (n = 423) | | Non-CVD (n = 4,592) | | P value |
|---|---|---|---|---|---|---|---|
| | Mean | ± SD | Mean | ± SD | Mean | ± SD | |
| | N | % | N | % | N | % | |
| Age | 65.8 | ± 7.6 | 71.2 | ± 8.9 | 65.4 | 7.2 | <0.001 |
| Sex | | | | | | | <0.001 |
| Male | 1747 | 34.8 | 203 | 48.0 | 1544 | 33.6 | |
| Female | 3268 | 65.2 | 220 | 52.0 | 3048 | 66.4 | |
| Race | | | | | | | <0.001 |
| Chinese | 4541 | 90.6 | 354 | 83.9 | 4187 | 91.2 | |
| Others | 470 | 9.4 | 68 | 16.1 | 402 | 8.8 | |
| Education | | | | | | | <0.001 |
| No education | 953 | 19.0 | 137 | 32.4 | 816 | 17.8 | |
| Primary (1–6 years) | 1916 | 38.3 | 154 | 36.4 | 1762 | 38.4 | |
| Post-primary (> 6 years) | 2139 | 42.7 | 132 | 31.2 | 2007 | 43.8 | |
| Housing type | | | | | | | <0.001 |
| 1–2 room | 669 | 13.4 | 96 | 22.7 | 573 | 12.5 | |
| 3–5 room | 3408 | 68.1 | 256 | 60.7 | 3152 | 68.8 | |
| High end public/private housing | 924 | 18.5 | 70 | 16.6 | 854 | 18.6 | |
| Single, divorced or widowed | 1519 | 30.3 | 170 | 40.2 | 1349 | 29.4 | <0.001 |
| Living alone | 569 | 11.4 | 61 | 14.5 | 508 | 11.1 | 0.038 |
| Current smoking | 402 | 8.03 | 55 | 13.0 | 347 | 7.6 | <0.001 |
| Alcohol drinking | 178 | 3.6 | 26 | 6.2 | 152 | 3.3 | 0.003 |
| Frailty status | | | | | | | <0.001 |
| Robust | 2513 | 50.1 | 157 | 37.1 | 2356 | 51.3 | |
| Prefrail | 2317 | 46.2 | 228 | 53.9 | 2089 | 45.5 | |
| Frail | 185 | 3.7 | 38 | 9.0 | 147 | 3.2 | |
| Shrinking | 430 | 8.6 | 52 | 12.3 | 378 | 8.2 | 0.004 |
| Low activity | 1322 | 26.4 | 142 | 33.6 | 1180 | 25.7 | <0.001 |
| Weakness | 907 | 18.1 | 133 | 31.4 | 774 | 16.9 | <0.001 |
| Exhaustion | 572 | 11.4 | 64 | 15.1 | 508 | 11.1 | 0.012 |
| Slowness | 223 | 4.5 | 44 | 10.4 | 179 | 3.9 | <0.001 |
| Depressed (GDS>5) | 275 | 5.5 | 37 | 8.8 | 238 | 5.2 | 0.002 |
| Cognitive impaired (MMSE<24) | 480 | 9.6 | 104 | 24.6 | 376 | 8.2 | <0.001 |
| Metabolic syndrome | 1281 | 25.5 | 125 | 29.5 | 1156 | 25.2 | 0.048 |
| Central obesity | 2519 | 50.3 | 210 | 49.6 | 2309 | 50.3 | 0.825 |
| Raised TG (>1.7) | 1348 | 26.9 | 140 | 33.1 | 1208 | 26.3 | 0.003 |
| Reduced HDL-C | 1179 | 23.5 | 112 | 26.5 | 1067 | 23.2 | 0.132 |
| Raised LDL-C | 2140 | 45.0 | 184 | 45.8 | 1956 | 45.0 | 0.752 |
| Hypertension | 3638 | 72.5 | 369 | 87.2 | 3269 | 71.2 | <0.001 |
| Diabetes | 1441 | 28.7 | 169 | 39.9 | 1272 | 27.7 | <0.001 |
| Haemoglobin (g/dL) | 13.4 | ± 1.4 | 13.4 | ± 1.6 | 13.4 | ± 1.4 | 0.589 |
| Albumin (g/L) | 42.3 | ± 2.9 | 41.5 | ± 3.2 | 42.3 | ± 2.8 | <0.001 |
| Creatinine (umol/L) | 73.8 | ± 34.1 | 89.9 | ± 58.4 | 72.3 | ± 30.5 | <0.001 |
| White blood cell (x10^9/L) | 6.0 | ± 1.6 | 6.3 | ± 1.8 | 6.0 | ± 1.6 | <0.001 |
| Statin therapy | 1251 | 25.0 | 103 | 24.4 | 1148 | 25 | 0.767 |
| Antiplatelet therapy | 186 | 3.7 | 30 | 7.1 | 156 | 3.4 | <0.001 |

(*Continued*)

**Table 1.** (Continued)

| Baseline characteristics | Whole sample (n = 5015) | | CVD (n = 423) | | Non-CVD (n = 4,592) | | P value |
|---|---|---|---|---|---|---|---|
| | Mean | ± SD | Mean | ± SD | Mean | ± SD | |
| | N | % | N | % | N | % | |
| Anticoagulant therapy | 8 | 0.2 | 1 | 0.2 | 7 | 0.2 | 0.679 |

Abbreviations: CVD, cardiovascular disease; GDS, Geriatric Depression Scale; HDL-C, high-density lipoprotein cholesterol; LDL-C, low-density lipoprotein cholesterol; MMSE, Mini-Mental State Examination; TG, triglycerides.

metabolic, vascular and inflammatory risk factors or markers: diabetes, hypertension, dyslipidemia, metabolic syndrome, as well as low albumin, high creatinine and white cell count.

We observed 423 CVD events from a total of 51,135.2 person-years (p-y) of follow-up observation; overall CVD incidence rate (IR): 8.3 per 100,000 p-y. Among 423 CVD cases, 155 were non-fatal MI, 164 were non-fatal strokes and 104 were fatal CVD. The estimated risks of CVD events overall from follow up observation according to baseline categories of robust, pre-frail and frail participants are shown in Table 2.

## Overall CVD

Compared to robust individuals, pre-frail and frail individuals were more likely to show higher risks of overall CVD. Adjusted for age, sex, education and housing type, pre-frailty-associated HR = 1.26 (95% CI: 1.02–1.56), frailty-associated HR = 1.82, (95% CI: 1.24–2.66) (Table 3). Including additional model covariates of vascular and cardio-metabolic risk factors resulted in

**Table 2. Follow-up incidence rate of CVD events by baseline frailty status.**

| | Incident event N | Person-years (p-y) of observation | Incidence /1,000 p-y | (95% CI) |
|---|---|---|---|---|
| Overall CVD | | | | |
| Robust | 157 | 26336.2 | 6.0 | (5.08, 6.95) |
| Prefrail | 228 | 23262.7 | 9.8 | (8.59, 11.14) |
| Frail | 38 | 1520.7 | 25.0 | (17.90, 33.90) |
| Non-fatal MI | | | | |
| Robust | 61 | 25870.4 | 2.4 | (1.82, 3.01) |
| Prefrail | 84 | 22334.9 | 3.8 | (3.02, 4.63) |
| Frail | 10 | 1310.2 | 7.6 | (3.88, 13.61) |
| Non-fatal stroke | | | | |
| Robust | 69 | 25796.8 | 2.7 | (2.10, 3.36) |
| Prefrail | 85 | 22253.6 | 3.8 | (3.07, 4.70) |
| Frail | 10 | 1289.5 | 7.8 | (3.94, 13.83) |
| Non-fatal CVD | | | | |
| Robust | 130 | 25591.5 | 5.1 | (4.26, 6.01) |
| Prefrail | 169 | 21998.9 | 7.7 | (6.59, 8.91) |
| Frail | 20 | 1271.7 | 15.7 | (9.87, 23.85) |
| Fatal CVD | | | | |
| Robust | 27 | 26082.5 | 1.0 | (0.69, 1.48) |
| Prefrail | 59 | 22626.5 | 2.6 | (2.00, 3.34) |
| Frail | 18 | 1331.1 | 13.5 | (8.27, 20.96) |

Abbreviations: CVD, cardiovascular disease; MI, Myocardial Infarction; CI, confidence interval.

**Table 3. Associations between frailty status at baseline and incidence of CVD events.**

| | Unadjusted model | | Model 1 | | Model 2 | | Model 3 | | Model 4 | | Model 5 | |
|---|---|---|---|---|---|---|---|---|---|---|---|---|
| | SHR (95%CI) | P value | SHR (95%CI) | P value | SHR (95%CI) | P value | SHR (95%CI) | P value | SHR (95%CI) | P value | SHR (95%CI) | P value |
| **Overall CVD** | | | | | | | | | | | | |
| Robust | 1 (Reference) | | 1 (Reference) | | 1 (Reference) | | 1 (Reference) | | 1 (Reference) | | 1 (Reference) | |
| Prefrail | 1.65 (1.35–2.02) | <0.001 | 1.36 (1.11–1.68) | 0.004 | 1.26 (1.02–1.56) | 0.031 | 1.26 (1.01–1.57) | 0.038 | 1.24 (0.99–1.55) | 0.053 | 1.22 (0.98–1.53) | 0.074 |
| Frail | 4.31 (3.02–6.15) | <0.001 | 2.02 (1.38–2.96) | <0.001 | 1.82 (1.24–2.66) | 0.002 | 1.54 (1.00–2.35) | 0.047 | 1.35 (0.87–2.09) | 0.184 | 1.30 (0.84–2.03) | 0.243 |
| **Non-fatal MI** | | | | | | | | | | | | |
| Robust | 1 (Reference) | | 1 (Reference) | | 1 (Reference) | | 1 (Reference) | | 1 (Reference) | | 1 (Reference) | |
| Prefrail | 1.57 (1.13–2.19) | 0.007 | 1.37 (0.98–1.91) | 0.069 | 1.20 (0.86–1.70) | 0.286 | 1.23 (0.87–1.75) | 0.245 | 1.19 (0.84–1.70) | 0.334 | 1.16 (0.81–1.67) | 0.407 |
| Frail | 2.97 (1.51–5.82) | 0.002 | 1.52 (0.73–3.18) | 0.263 | 1.21 (0.59–2.48) | 0.594 | 0.91 (0.40–2.07) | 0.816 | 0.81 (0.35–1.91) | 0.636 | 0.77 (0.32–1.86) | 0.568 |
| **Non-fatal stroke** | | | | | | | | | | | | |
| Robust | 1 (Reference) | | 1 (Reference) | | 1 (Reference) | | 1 (Reference) | | 1 (Reference) | | 1 (Reference) | |
| Prefrail | 1.47 (1.08–2.02) | 0.016 | 1.28 (0.93–1.78) | 0.135 | 1.27 (0.92–1.77) | 0.147 | 1.23 (0.87–1.75) | 0.243 | 1.25 (0.88–1.78) | 0.213 | 1.25 (0.87–1.79) | 0.224 |
| Frail | 3.08 (1.56–6.06) | 0.001 | 1.55 (0.74–3.26) | 0.247 | 1.48 (0.70–3.10) | 0.304 | 1.50 (0.68–3.31) | 0.313 | 1.34 (0.57–3.13) | 0.497 | 1.43 (0.61–3.38) | 0.414 |
| **Non-fatal CVD** | | | | | | | | | | | | |
| Robust | 1 (Reference) | | 1 (Reference) | | 1 (Reference) | | 1 (Reference) | | 1 (Reference) | | 1 (Reference) | |
| Prefrail | 1.50 (1.19–1.88) | <0.001 | 1.30 (1.03–1.64) | 0.027 | 1.21 (0.96–1.54) | 0.112 | 1.21 (0.94–1.54) | 0.135 | 1.18 (0.92–1.52) | 0.182 | 1.18 (0.92–1.52) | 0.200 |
| Frail | 2.91 (1.80–4.70) | <0.001 | 1.49 (0.88–2.53) | 0.135 | 1.32 (0.79–2.20) | 0.292 | 1.11 (0.63–1.96) | 0.721 | 0.94 (0.52–1.73) | 0.851 | 0.93 (0.50–1.73) | 0.825 |
| **Fatal CVD** | | | | | | | | | | | | |
| Robust | 1 (Reference) | | 1 (Reference) | | 1 (Reference) | | 1 (Reference) | | 1 (Reference) | | 1 (Reference) | |
| Prefrail | 2.53 (1.61–3.99) | <0.001 | 1.91 (1.21–3.04) | 0.006 | 1.83 (1.16–2.89) | 0.009 | 1.79 (1.12–2.88) | 0.016 | 1.76 (1.09–2.84) | 0.021 | 1.70 (1.05–2.77) | 0.032 |
| Frail | 13.5 (7.43–24.4) | <0.001 | 4.50 (2.30–8.83) | <0.001 | 3.88 (2.00–7.50) | <0.001 | 3.05 (1.49–6.27) | 0.002 | 2.88 (1.38–6.00) | 0.005 | 2.48 (1.14–5.37) | 0.021 |

Abbreviations: CVD, cardiovascular disease; MI, Myocardial Infarction; CI, confidence interval.

Model 1: Adjusted for age, sex.

Model 2: Adjusted for Model 1 plus ethnicity, education, housing.

Model 3: Adjusted for Model 2 plus smoking, alcohol, central obesity, raised triglycerides, reduced high-density lipoprotein cholesterol, hypertension, diabetes, raised low-density lipoprotein cholesterol, statin therapy, antiplatelet therapy, anticoagulant therapy.

Model 4: Adjusted for Model 3 plus depression by Geriatric Depression Scale, cognitive impairment by Mini-Mental State Examination.

Model 5: Adjusted for Model 4 plus blood biomarkers (albumin, haemoglobin, white blood cell, creatinine).

no substantial alteration in the HR estimates: pre-frailty-associated HR = 1.26 (95% CI: 1.01–1.57), frailty-associated HR = 1.54 (95% CI: 1.00–2.35).

Additional covariates of MMSE and GDS depression resulted in non-significant estimates of pre-frailty associated HR = 1.24 (95% CI: 0.99–1.55), frailty-associated HR = 1.35 (95% CI: 0.87–2.09). Additional covariates of blood biomarkers (albumin, creatinine, WBC, haemoglobin) resulted in further reduced and non-significant estimates of pre-frailty-associated HR = 1.22 (95% CI: 0.98–1.53), frailty-associated HR = 1.30, (95% CI: 0.84–2.03).

## Fatal CVD events

Consistent and robust estimates of association in all models were observed for fatal CVD. In the full model with all covariates (Table 3, Model 5), significant estimates remained: SHR = 1.70 (95% CI: 1.05–2.77) in prefrail group and SHR = 2.48 (95% CI: 1.14–5.37) in frail group.

*Non-fatal CVD* (including acute MI and stroke) rates were higher in pre-frail and frail individuals, based on small sample sizes, and the covariate-adjusted SHR were not statistically significant in Model 2, and not shown for additional covariate adjustments.

For individual components of frailty, significant associations in Model 4 (Table 4) were seen for weakness (HR = 1.36, 95% CI: 1.07–1.72), and shrinking (HR = 1.39, 95%CI: 1.01–1.89) with overall CVD events. Weakness, shrinking, and exhaustion showed significant associations with fatal CVD. There were no significant associations among non-fatal CVD outcomes.

There was a total of 692 all-cause deaths over 50040.1 patient-years at risk, including 228 robust, 387 prefrail, and 77 frail participants. Compared with robust participants, prefrail and frail participants were both associated with higher risk of all-cause mortality. In the unadjusted model, increased risk for all-cause mortality was observed in both prefrail (vs robust, HR = 1.99, 95% CI:1.69–2.35, p<0.001) and frail (vs robust, HR = 7.49, 95% CI:5.78–9.72, p < 0.001). This significant association was consistent across all models with pre-frailty-associated HR = 1.40 (95% CI: 1.17–1.67, p<0.001), frailty-associated HR = 2.03 (95% CI:1.48–2.80, p<0.001) in Model 5 when all the covariate adjusted. All five frailty components except low activity showed significant associations with overall mortality: shrinking (HR = 1.51, 95% CI:1.19–1.91, p = 0.001), weakness (HR = 1.62, 95% CI:1.35–1.94, p<0.001), exhaustion (HR = 1.29, 95% CI:1.03–1.61, p = 0.028), slowness (HR = 1.53, 95% CI:1.16–2.02, p = 0.003).

In further sensitivity analyses, we excluded CVD cases within 1 year after baseline and found similar results.

**Table 4. Association between frailty components at baseline and incidence of CVD events in the follow-up.**

| Frailty components | HR (95% CI) | P value |
|---|---|---|
| **Overall CVD** | | |
| Shrinking | 1.39 (1.01–1.89) | 0.041 |
| Low activity | 1.02 (0.82–1.27) | 0.848 |
| Weakness | 1.36 (1.07–1.72) | 0.011 |
| Exhaustion | 1.12 (0.83–1.50) | 0.467 |
| Slowness | 1.01 (0.69–1.48) | 0.968 |
| **Fatal CVD** | | |
| Shrinking | 1.98 (1.12–3.49) | 0.018 |
| Low activity | 1.02 (0.64–1.62) | 0.939 |
| Weakness | 1.79 (1.13–2.84) | 0.013 |
| Exhaustion | 2.32 (1.44–3.74) | 0.001 |
| Slowness | 1.54 (0.82–2.90) | 0.182 |

Abbreviations: CVD, cardiovascular disease; HR, hazard ratio; CI, confidence interval.

HR adjusted for age, sex, ethnicity, education, housing, smoking, alcohol, central obesity, raised triglycerides, reduced high-density lipoprotein cholesterol, hypertension, diabetes, raised low-density lipoprotein cholesterol, statin therapy, antiplatelet therapy, anticoagulant therapy, depression by Geriatric Depression Scale, cognitive impairment by Mini-Mental State Examination.

## Discussion

Our study, in agreement with previous studies showed that pre-frailty and frailty were associated with increased risks of overall CVD events [24], and frailty status was a significant predictor of all-cause mortality [4, 25]. However, previous studies have not reported the separate risks of non-fatal and fatal CVD events and did not control for the effects of depression. We observed in this study that pre-frailty and frailty were significantly associated with 1.3 and 1.7-fold increased risk of CVD overall, adjusted for sociodemographic, behavioral and cardio-metabolic and vascular risk factors, but not with subsequent adjustment for depression and cognitive impairment and blood biomarkers. However, pre-frailty and frailty were robustly associated respectively with 1.6-fold and 2.6-fold increased risk of fatal CVD in the fully adjusted model, whereas no significant associations were found for risk of non-fatal CVD events (acute MI or stroke).

Our study may provide clues to the mechanistic and developmental relationship by showing significant findings in the stepwise analysis after adjustment of traditional cardio-metabolic and vascular risk factors, medication therapies, depression, cognitive factors, and biomarkers. The results suggest that frailty clearly has a powerful influence in increasing the risk of dying from cardiovascular disease. Its significant HR after adjustment for cardio-metabolic and vascular risk factors was attenuated after adjustment for depression, cognitive impairment, and surrogate blood markers of chronic inflammation. This suggests that comorbid depression, and associated cognitive impairment and chronic inflammation, contributes to the increased CVD mortality risk among pre-frail and frail individuals.

We found that frailty was associated with a moderate (less than 50%) increased risk of non-fatal CVD incidence after adjustment for traditional CVD risk factors. This suggests that the role of frailty per se in promoting the development and clinical onset of cardiovascular disease is relatively subtle. Previous research showed that the presence of frailty among MI patients was significantly associated with increased CVD mortality [26, 27]. The population study by Veronese et al. 2017 [14] controlled for the presence of carotid intima media thickness, presence of carotid plaque and total coronary calcifications, and found that frailty (HR = 1.35; 95% CI: 1.05–1.74) remained significantly associated with CVD events overall, indicating that in the presence of subclinical atherosclerotic disease, it is an independent CVD risk factor. More prospective studies are needed to elucidate the longitudinal relationships between frailty measures and preclinical cardiovascular disease.

It thus appears that the frailty syndrome has a complex mechanistic link with the development of incipient CVD and with its final progression to fatal outcome. It is possible that frailty precipitates clinically overt CVD and/or accelerates disease progression from baseline subclinical atherosclerotic disease. The metabolic syndrome cluster of cardio-metabolic risk factors is well known to predict higher CVD [28] and stroke [29] risks, and has also been found to be associated with increased risk of incident frailty [30–32]. Independent inverse associations between subclincial measures of arterial disease with muscle mass and functional decline have also been reported in some studies [33, 34], but not in others [35].

Among component measures of frailty in this study, weakness showed significant association with the increased risk of overall CVD, fatal CVD, as well as all-cause mortality, which was in line with previously studies [24, 36–38]. As weakness was assessed by knee extension strength or POMA battery, which are both objective measurements for muscle strength, its strong predicting value for higher risk of CVD in our study suggested that preventions on muscle strength decline may potentially reduce the risk of CVD and mortality for older adults. Consistent with other studies [38, 39], slowness also presented higher risk of all-cause mortality in our study. However, we failed to find significant association between slowness and risk

of CVD after adjusting for traditional CVD risk factors and medication therapies. Although study conducted by Veronese et al. [14] showed similar findings, some other studies [9, 13] concluded slow gait speed was a significant predictor for CVD. This inconsistency may be due to the different measurements and cutoffs for slowness definitions.

In this follow-up population without overt CVD at baseline, the prevalence of frailty (46%) and frailty (3.7%) is very high, but this is not exceptional, as it has been reported in many studies worldwide. Pre-frailty is a transitional precursor state of frailty, and both are reversible by multi-domain lifestyle and health interventions (nutritional, physical, cognitive interventions, polypharmacy de-prescription, vitamin D supplementation) [17, 18]. Further interventional studies should be conducted to evaluate the potential benefits of pre-frailty and frailty interventions to reduce the risk of CVD and mortality risks.

In this large prospective cohort study of community-dwelling middle-aged and older adults in an Asian population, case ascertainment of CVD events using computerized record linkage with the national registry of disease was accurate and complete. The sample was however still underpowered to detect significant associations for non-fatal CVD and especially stroke. A limitation is that non-fatal CVD included only acute MI and stroke and deaths from CVD included heart failure, but non-fatal cases of heart failure from hospitalization records were not ascertained. Another limitation is that low haemoglobin, low albumin, and white blood cell counts are non-specific indirect measures of inflammation, more specific established markers such as IL6 or TNF-alpha were not employed. Additionally, due to the small case number of fatal stroke and fatal MI, we were unable to further explore the relationship between frailty and the risk of fatal stroke and/or fatal MI specifically.

## Conclusions

We demonstrated that pre-frailty and frailty were significantly associated with increased risks of incident CVD, and fatal CVD in particular. Given that they are reversible by early intervention, there are potential benefits in reducing CVD burden and mortality from interventions targeting pre-frailty and early frailty that should be further investigated in future clinical studies.

## Acknowledgments

We thank the following voluntary welfare organizations for their support: Geylang East Home for the Aged, Presbyterian Community Services, St Luke's Eldercare Services, Thye Hua Kwan Moral Society (Moral Neighbourhood Links), Yuhua Neighbourhood Link, Henderson Senior Citizens' Home, NTUC Eldercare Co-op Ltd, Thong Kheng Seniors Activity Centre (Queenstown Centre) and Redhill Moral Seniors Activity Centre. We thank National Registry of Disease Office (NRDO) for performing the computerized record linkage and providing access to the National Disease Registry data.

## Author Contributions

**Conceptualization:** Tze Pin Ng.

**Data curation:** Xiao Liu, Qi Gao, Xinyi Gwee.

**Formal analysis:** Xiao Liu, Nien Xiang Tou.

**Funding acquisition:** Tze Pin Ng.

**Investigation:** Xiao Liu, Tze Pin Ng.

**Methodology:** Xiao Liu, Tze Pin Ng.

**Supervision:** Tze Pin Ng.

**Visualization:** Xiao Liu.

**Writing – original draft:** Xiao Liu.

**Writing – review & editing:** Xiao Liu, Nien Xiang Tou, Qi Gao, Xinyi Gwee, Shiou Liang Wee, Tze Pin Ng.

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
