## [Decision Letter · Decision Letter 0]

30 Dec 2021

PONE-D-21-35749Frailty and Risk of Cardiovascular Disease and MortalityPLOS ONE

Dear Dr. Tze Pin Ng,

Thank you for submitting your manuscript to PLOS ONE. After careful consideration, we feel that it has merit but does not fully meet PLOS ONE’s publication criteria as it currently stands. Therefore, we invite you to submit a revised version of the manuscript that addresses the points raised during the review process.

The manuscript is necessary to be revised according to the two reviewes' comments.

We look forward to receiving your revised manuscript.

Kind regards,

Masaki Mogi

Academic Editor

PLOS ONE

Journal Requirements:

Reviewers' comments:

Reviewer's Responses to Questions

**Comments to the Author**

1. Is the manuscript technically sound, and do the data support the conclusions?

Reviewer #1: Yes

Reviewer #2: Partly

2. Has the statistical analysis been performed appropriately and rigorously? 

Reviewer #1: Yes

Reviewer #2: I Don't Know

3. Have the authors made all data underlying the findings in their manuscript fully available?

Reviewer #1: Yes

Reviewer #2: No

4. Is the manuscript presented in an intelligible fashion and written in standard English?

Reviewer #1: Yes

Reviewer #2: Yes

5. Review Comments to the Author

Reviewer #1: The authors examined the relationship between frailty and CVD events in prospective study. It is interesting topics and outcomes.

I have several comments.

It is interesting that frailty is associated with the occurrence of fatal CVD events. Which of the MI or stroke leads to fatal conditions? MI? Stroke? or both? Please add the results of examination the relationship between (1) frailty and fatal MI, and (2) frailty and fatal stroke.

Did the medication therapy such as antiplatelet therapy, anticoagulant therapy, and statin therapy influence the occurrence of CVD events?

I think that cognitive impairment is one of the important factor associated with frailty. Did the cognitive impairment or depression affect the CVD events including non-fatal MI, non-fatal stroke, fatal MI, fatal stroke, and fatal CVD?

I’m interested in the relationship between frailty and overall survival in your study population. Did the prefrail and frail affect overall mortality?

Reviewer #2: The authors have reported an association of pre-frailty and frailty with the risk of developing cardiovascular disease (CVD) morbidity and mortality over 10 years in a prospective cohort study of community-dwelling older adults in an Asian population. Although it is interesting, the present paper has several issues to be resolved as below.

1. What was the breakdown of cardiovascular diseases that were defined in this study? In particular, that of non-fatal cardiovascular diseases are unclear.

2. In Table 1, why raised or reduced low-density lipoprotein (LDL-C) was not shown? Isn’t the serum LDL-C a risk factor of CVD? Should several statistical analyses be adjusted for the serum levels of LDL-C? Moreover, the present study has lacked the statistical analysis adjusted for the medication history, especially for statin, antihypertensive and diabetes drugs.

3. Regarding the study subjects of 5,015, please show detail data of the follow-up periods of them.

4. In DISCUSSION, the authors have described as below: “Our study sheds light on the mechanistic and developmental relationship between… (page 16, line 1)”. However, the present study has only shown a relationship between pre-frailty/frailty and CVD outcomes. In addition, the frailty measurements to assess pre-frailty/frailty comprehensively was actually composed of unquantifiable measurement items of “Exhaustion” and “Low activity”. Isn’t the above description by the authors overestimated?

6. PLOS authors have the option to publish the peer review history of their article (what does this mean?). If published, this will include your full peer review and any attached files.

Reviewer #1: No

Reviewer #2: No

---

## [Author Response · Author response to Decision Letter 0]

15 Jun 2022

Dear Reviewers,

Thank you very much for giving us the opportunity to submit a revised draft of our manuscript. We greatly appreciate the time and effort that you have dedicated to providing your valuable feedback and insightful comments on our manuscript. To address all the concerns raised, we have merged new variables in our dataset, redone our analysis, and revised our manuscript accordingly. We’ve tracked the changes in the manuscript. 

Here is our point-by-point response to your comments and concerns. All the line numbers are based on “All Markup” viewing version.

Comments from Reviewer 1

Reviewer #1: The authors examined the relationship between frailty and CVD events in prospective study. It is interesting topics and outcomes. I have several comments.

1. It is interesting that frailty is associated with the occurrence of fatal CVD events. Which of the MI or stroke leads to fatal conditions? MI? Stroke? or both? Please add the results of examination the relationship between (1) frailty and fatal MI, and (2) frailty and fatal stroke.

Reply:

Thank you for pointing out that it would be interesting to explore the relationship specifically between frailty and fatal MI and/or stroke. Unfortunately, the number of fatal MI/stroke cases in our dataset was too small to perform the analysis: total number of fatal MI was 32 with only 1 frail case; and total number of fatal strokes was 10, all of which were prefrail cases. In this case, we added one more limitation in our discussion in Line 790-792:

 “Additionally, due to the small case number of fatal stroke and fatal MI, we were unable to further explore the relationship between frailty and the risk of fatal stroke and/or fatal MI specifically.” 

2. Did the medication therapy such as antiplatelet therapy, anticoagulant therapy, and statin therapy influence the occurrence of CVD events?

Reply:

Yes, we agree with this and we performed reanalysis by including antiplatelet therapy, anticoagulant therapy, and statin therapy into our dataset. Those covariates were added in model 3, model 4, and model 5. Our main findings remained the same after the re-analysis.

The method section had been revised accordingly in Line 189-190:

“Medication therapies included statin therapy, antiplatelet therapy, anticoagulant therapy.” 

And Line 206-208:

“Model 3: additionally for smoking, alcohol, central obesity, raised TG, reduced HDL-C, diabetes, hypertension, raised LDL-C, statin therapy, antiplatelet therapy, anticoagulant therapy; ”

All the relevant results had been updated for Table 1, Table 3, Table 4, Abstract, and Results section.

3. I think that cognitive impairment is one of the important factor associated with frailty. Did the cognitive impairment or depression affect the CVD events including non-fatal MI, non-fatal stroke, fatal MI, fatal stroke, and fatal CVD?

Reply:

In our first manuscript version, we had already adjusted depression by Geriatric Depression Scale, and cognitive impairment by Mini-Mental State Examination in Model 4 (Table3) which showed significant results for fatal CVD but not non-fatal cases. Now in this updated version, after adding the new covariates mentioned above, the results remained similar (Table 3 Model 4). 

4. I’m interested in the relationship between frailty and overall survival in your study population. Did the prefrail and frail affect overall mortality?

Reply:

Thank you for this suggestion. In our study, both prefrail and frail are significantly associated with higher risk of overall mortality. 

We have included the method/analysis section about overall mortality in Line 197-199:

“Hierarchical adjusted Cox proportional hazard models were used to estimate hazard ratios (HR) and their 95% confidence intervals (95% CIs) between frailty status and overall incident CVD, and between frailty status and overall mortality.”

 We added the results in the manuscript Line 626-636:

“There was a total of 692 all-cause deaths over 50040.1 patient-years at risk, including 228 robust, 387 prefrail, and 77 frail participants. Compared with robust participants, prefrail and frail participants were both associated with higher risk of all-cause mortality. In the unadjusted model, increased risk for all-cause mortality was observed in both prefrail (vs robust, HR = 1.99, 95% CI:1.69-2.35, p<0.001) and frail (vs robust, HR = 7.49, 95% CI:5.78-9.72, p < 0.001). This significant association was consistent across all models with pre-frailty-associated HR=1.40 (95% CI: 1.17-1.67, p<0.001), frailty-associated HR=2.03 (95% CI:1.48-2.80, p<0.001) in Model 5 when all the covariate adjusted. All five frailty components except low activity showed significant associations with overall mortality: shrinking (HR = 1.51, 95% CI:1.19-1.91, p=0.001), weakness (HR = 1.62, 95% CI:1.35-1.94, p<0.001), exhaustion (HR = 1.29, 95% CI:1.03-1.61, p=0.028), slowness (HR = 1.53, 95% CI:1.16-2.02, p=0.003).” 

We added relevant discussion in Line 703-705:

“Our study, in agreement with previous studies showed that pre-frailty and frailty were associated with increased risks of overall CVD events[24], and frailty status was a significant predictor of all-cause mortality[4, 25]”

Comments from Reviewer 2

Reviewer #2: The authors have reported an association of pre-frailty and frailty with the risk of developing cardiovascular disease (CVD) morbidity and mortality over 10 years in a prospective cohort study of community-dwelling older adults in an Asian population. Although it is interesting, the present paper has several issues to be resolved as below.

1. What was the breakdown of cardiovascular diseases that were defined in this study? In particular, that of non-fatal cardiovascular diseases are unclear.

Reply:

Thank you for pointing this out. Non-fatal CVD cases included both non-fatal MI and non-fatal stroke which obtained from Singapore government registry data. 

We had revised our method part to clarify the relevant definitions in Line 133-141:

“All-cause mortality and fatal CVD cases were obtained from the Death Registry data from Singapore National Registry of Diseases Office based on International Classification of Diseases (ICD). Fatal CVDs were identified using ICD 9 codes from 390 to 459 or ICD 10 codes from I00 to I99. Other CVD outcomes included 1) non-fatal MI, obtained from Singapore Myocardial Infarction Registry; 2) non-fatal stroke, obtained from Singapore Stroke Registry; 3) non-fatal CVD, defined as an inclusion of non-fatal MI and non-fatal stroke. Overall CVD included both fatal CVD and non-fatal CVD. Overall mortality includes all-cause of death cases.”

2. In Table 1, why raised or reduced low-density lipoprotein (LDL-C) was not shown? Isn’t the serum LDL-C a risk factor of CVD? Should several statistical analyses be adjusted for the serum levels of LDL-C? 

Moreover, the present study has lacked the statistical analysis adjusted for the medication history, especially for statin, antihypertensive and diabetes drugs.

Reply:

Thank you for this suggestion. We agree that LDL-C and medical history should be adjusted in our analysis. We had redone the analysis by adding LDL-C and medication therapy (antiplatelet therapy, anticoagulant therapy, and statin therapy) in Model 3, Model 4, and Model 5. Our main findings remained the same after the re-analysis.

The method section had been revised accordingly in Line 188-190

“Raised low-density lipoprotein cholesterol (LDL-C) was defined as ≥3.4mmol/l[23]. Medication therapies included statin therapy, antiplatelet therapy, anticoagulant therapy.” 

And Line 206-208:

“Model 3: additionally for smoking, alcohol, central obesity, raised TG, reduced HDL-C, diabetes, hypertension, raised LDL-C, statin therapy, antiplatelet therapy, anticoagulant therapy;”

All the relevant results had been updated for Table 1, Table 3, Table 4, Abstract, and Results section.

3. Regarding the study subjects of 5,015, please show detail data of the follow-up periods of them.

Reply:

We clarified the follow-up period in Line 139-141. 

“The follow-up time for this study started at the date of participants enrolment and ended in December 2017 for all the outcomes.”

Participants’ enrollment time can be found in Line 118-121:

“SLAS-1 recruited 2,800 older persons in the South-East Region in 2003-2004, and SLAS-2 recruited 3,200 individuals in the South Central and Western Region in Singapore in 2009-2013, each with 3 to 5 yearly follow-ups.”

4. In DISCUSSION, the authors have described as below: “Our study sheds light on the mechanistic and developmental relationship between… (page 16, line 1)”. However, the present study has only shown a relationship between pre-frailty/frailty and CVD outcomes. 

Reply:

Thank you for this comment. Yes, we investigated the relationship between frailty status and CVD but the results of stepwise analysis provided some clues on the mechanistic behind this relationship. 

To make it clear, we had revised the manuscript in Line 714-716:

“Our study provides clues to the mechanistic and developmental relationship by showing significant findings in the stepwise analysis after adjustment of traditional cardio-metabolic and vascular risk factors, medication therapies, depression, cognitive factors, and biomarkers.”

In addition, the frailty measurements to assess pre-frailty/frailty comprehensively was actually composed of unquantifiable measurement items of “Exhaustion” and “Low activity”. Isn’t the above description by the authors overestimated?

Reply:

We acknowledge that “exhaustion” and “low activity” were self-reported measurement that may not be as accurate as other quantifiable objective measurements. Both measurements were integral components of the well-established Fried phenotype and defined accordingly. In this regard, we revise our manuscript by focusing our discussion on those objective measurements in Line 749-769.

“Among component measures of frailty in this study, weakness showed significant association with the increased risk of overall CVD, fatal CVD, as well as all-cause mortality, which was in line with previously studies[24, 36-38]. As weakness was assessed by knee extension strength or POMA battery, which are both objective measurements for muscle strength, its strong predicting value for higher risk of CVD in our study suggested that preventions on muscle strength decline may potentially reduce the risk of CVD and mortality for older adults. Consistent with other studies[38, 39], slowness also presented higher risk of all-cause mortality in our study. However, we failed to find significant association between slowness and risk of CVD after adjusting for traditional CVD risk factors and medication therapies. Although a study conducted by Veronese et al[14] showed similar findings, some other studies[9, 13] concluded slow gait speed was a significant predictor for CVD. This inconsistency may be due to the different measurements and cutoffs for slowness definitions.”

Sincerely,

Authors from manuscript [PONE-D-21-35749]

---

## [Decision Letter · Decision Letter 1]

4 Jul 2022

PONE-D-21-35749R1Frailty and Risk of Cardiovascular Disease and MortalityPLOS ONE

Dear Dr. Ng,

Thank you for submitting your manuscript to PLOS ONE. After careful consideration, we feel that it has merit but does not fully meet PLOS ONE’s publication criteria as it currently stands. Therefore, we invite you to submit a revised version of the manuscript that addresses the points raised during the review process.

Minor revisions are necessary in the revised version. See and repsond the comments.

We look forward to receiving your revised manuscript.

Kind regards,

Masaki Mogi

Academic Editor

PLOS ONE

Journal Requirements:

Reviewers' comments:

Reviewer's Responses to Questions

**Comments to the Author**

1. If the authors have adequately addressed your comments raised in a previous round of review and you feel that this manuscript is now acceptable for publication, you may indicate that here to bypass the “Comments to the Author” section, enter your conflict of interest statement in the “Confidential to Editor” section, and submit your "Accept" recommendation.

Reviewer #1: All comments have been addressed

Reviewer #2: (No Response)

2. Is the manuscript technically sound, and do the data support the conclusions?

Reviewer #1: Yes

Reviewer #2: Partly

3. Has the statistical analysis been performed appropriately and rigorously? 

Reviewer #1: Yes

Reviewer #2: N/A

4. Have the authors made all data underlying the findings in their manuscript fully available?

Reviewer #1: Yes

Reviewer #2: Yes

5. Is the manuscript presented in an intelligible fashion and written in standard English?

Reviewer #1: Yes

Reviewer #2: Yes

6. Review Comments to the Author

Reviewer #1: The authors have addressed all the comments from the previous review.

I vote to accept this manuscript as authors have made significant changes to the questions posed to them.

Reviewer #2: Thank you for replies to my previous comments. Almost of those have been appropriately addressed by the authors; however, there is an issue to be revised in the text.

Although the authors have described in the term of DISCUSSION as follows “Our study provides clues to the mechanistic and development relationship by showing....”, it’s better to revise as follows “Our study may provide clues to the mechanistic and development relationship by showing....”.

7. PLOS authors have the option to publish the peer review history of their article (what does this mean?). If published, this will include your full peer review and any attached files.

Reviewer #1: No

Reviewer #2: No

---

## [Author Response · Author response to Decision Letter 1]

20 Jul 2022

Dear Reviewers,

Thank you very much for taking your time to review our manuscript and for providing us with valuable comments. Please see our reply below:

Reviewers' comments:

Reviewer's Responses to Questions

Comments to the Author

1. If the authors have adequately addressed your comments raised in a previous round of review and you feel that this manuscript is now acceptable for publication, you may indicate that here to bypass the “Comments to the Author” section, enter your conflict of interest statement in the “Confidential to Editor” section, and submit your "Accept" recommendation.

Reviewer #1: All comments have been addressed

Reviewer #2: (No Response)

Reply to Reviewer #2:

Thank you very much for all your comments. We’ve revised the manuscript according to your comments.

2. Is the manuscript technically sound, and do the data support the conclusions?

Reviewer #1: Yes

Reviewer #2: Partly

Reply to Reviewer #2:

We acknowledge that our manuscript may not be technically perfect. We have made every effort to minimize confounding bias in the data analysis. Our re-analysis of the data involves additional adjustment for potential confounding variables to make the interpretation of the results and conclusion reasonably sound. 

3. Has the statistical analysis been performed appropriately and rigorously?

Reviewer #1: Yes

Reviewer #2: N/A

4. Have the authors made all data underlying the findings in their manuscript fully available?

Reviewer #1: Yes

Reviewer #2: Yes

5. Is the manuscript presented in an intelligible fashion and written in standard English?

Reviewer #1: Yes

Reviewer #2: Yes

6. Review Comments to the Author

Reviewer #1: The authors have addressed all the comments from the previous review.

I vote to accept this manuscript as authors have made significant changes to the questions posed to them.

Reviewer #2: Thank you for replies to my previous comments. Almost of those have been appropriately addressed by the authors; however, there is an issue to be revised in the text.

Although the authors have described in the term of DISCUSSION as follows “Our study provides clues to the mechanistic and development relationship by showing....”, it’s better to revise as follows “Our study may provide clues to the mechanistic and development relationship by showing....”.

Reply to Reviewer #2:

Thank you very much for this suggestion. We’ve revised the manuscript accordingly on Page 17:

“Our study may provide clues to the mechanistic and development relationship by showing....”

7. PLOS authors have the option to publish the peer review history of their article (what does this mean?). If published, this will include your full peer review and any attached files.

Do you want your identity to be public for this peer review? For information about this choice, including consent withdrawal, please see our Privacy Policy.

Reviewer #1: No

Reviewer #2: No

---

## [Editor Report · Decision Letter 2]

21 Jul 2022

Frailty and Risk of Cardiovascular Disease and Mortality

PONE-D-21-35749R2

Dear Dr. Ng,

We’re pleased to inform you that your manuscript has been judged scientifically suitable for publication and will be formally accepted for publication once it meets all outstanding technical requirements.

Kind regards,

Masaki Mogi

Academic Editor

PLOS ONE
---

## [Editor Report · Acceptance letter]

9 Sep 2022

PONE-D-21-35749R2 

Frailty and Risk of Cardiovascular Disease and Mortality 

Dear Dr. Ng:

I'm pleased to inform you that your manuscript has been deemed suitable for publication in PLOS ONE. Congratulations! Your manuscript is now with our production department. 

Kind regards, 

on behalf of

Dr. Masaki Mogi 

Academic Editor

PLOS ONE